# Dosimetric Effects of Air Cavities for MRI-Guided Online Adaptive Radiation Therapy (MRgART) of Prostate Bed after Radical Prostatectomy

**DOI:** 10.3390/jcm11020364

**Published:** 2022-01-12

**Authors:** Jonathan Pham, Minsong Cao, Stephanie M. Yoon, Yu Gao, Amar U. Kishan, Yingli Yang

**Affiliations:** 1Physics and Biology in Medicine IDP, University of California, 650 Charles E Young Drive S, Los Angeles, CA 90095, USA; jonathanpham@mednet.ucla.edu (J.P.); minsongcao@mednet.ucla.edu (M.C.); 2Department of Radiation Oncology, University of California, 200 Medical Plaza Driveway, Los Angeles, CA 90095, USA; smyoon@mednet.ucla.edu (S.M.Y.); yugao@mednet.ucla.edu (Y.G.); aukishan@mednet.ucla.edu (A.U.K.)

**Keywords:** MR-guided radiation therapy (MRgRT), prostate cancer, prostate bed, air cavity, dose calculation, adaptive therapy, low-field

## Abstract

Purpose: To evaluate dosimetric impact of air cavities and their corresponding electron density correction for 0.35 tesla (T) Magnetic Resonance-guided Online Adaptive Radiation Therapy (MRgART) of prostate bed patients. Methods: Three 0.35 T MRgRT plans (anterior–posterior (AP) beam, AP–PA beams, and clinical intensity modulated radiation therapy (IMRT)) were generated on a prostate bed patient’s (Patient A) planning computed tomography (CT) with artificial rectal air cavities of various sizes (0–3 cm, 0.5 cm increments). Furthermore, two 0.35 T MRgART plans (‘Deformed’ and ‘Override’) were generated on a prostate bed patient’s (Patient B) daily magnetic resonance image (MRI) with artificial rectal air cavities of various sizes (0–3 cm, 0.5 cm increments) and on five prostate bed patient’s (Patient 1–5) daily MRIs (2 MRIs: Fraction A and B) with real air cavities. For each MRgART plan, daily MRI electron density map was obtained by deformable registration with simulation CT. In the ‘Deformed’ plan, a clinical IMRT plan is calculated on the daily MRI with electron density map obtained from deformable registration only. In the ‘Override’ plan, daily MRI and simulation CT air cavities are manually corrected and bulk assigned air and water density on the registered electron density map, respectively. Afterwards, the clinical IMRT plan is calculated. Results: For the MRgRT plans, AP and AP–PA plans’ rectum/rectal wall max dose increased with increasing air cavity size, where the 3 cm air cavity resulted in a 20%/17% and 13%/13% increase, relative to no air cavity, respectively. Clinical IMRT plan was robust to air cavity size, where dose change remained less than 1%. For the MRgART plans, daily MRI electron density maps, obtained from deformable registration with simulation CT, was unable to accurately produce electron densities reflecting the air cavities. However, for the artificial daily MRI air cavities, dosimetric change between ‘Deformed’ and ‘Override’ plan was small (<4%). Similarly, for the real daily MRI air cavities, clinical constraint changes between ‘Deformed’ and ‘Override’ plan was negligible and did not lead to change in clinical decision for adaptive planning except for two fractions. In these fractions, the ‘Override’ plan indicated that the bladder max dose and rectum V35.7 exceeded the constraint, while the ‘Deformed’ plan showed acceptable dose, although the absolute difference was only 0.3 Gy and 0.03 cc, respectively. Conclusion: Clinical 0.35 T IMRT prostate bed plans are dosimetrically robust to air cavities. MRgART air cavity electron density correction shows clinically insignificant change and is not warranted on low-field systems.

## 1. Introduction

MR-guided radiation therapy (MRgRT) systems combine a magnetic resonance imaging (MRI) scanner with a linear accelerator (LINAC) radiation therapy system. The integrated MRI scanner allows for radiation-free on-board imaging with superior soft tissue contrast as opposed to conventionally used x-ray-based on-board imaging. Furthermore, MRI serves as a great tool for accurate tumor and critical structure contouring, enabling tighter treatment margins for dose escalation (stereotactic body radiation therapy (SBRT)) and treatment toxicity reduction [1]. Additionally, on-board MRI allows for accurate daily patient setup, which can be further utilized for MR-guided online adaptive radiation therapy (MRgART), where treatment plans can be modified based on the patient’s daily anatomy [2]. Lastly, real-time cine MRIs can be acquired during treatment for soft-tissue-based gating [3].

MRgRT has allowed for more personalized treatment, but it has also introduced new treatment variables, not previously considered in conventional external beam radiation therapy (EBRT). In particular, the effects of the magnetic field on the radiation beam have been a primary concern for patients being treated on MRgRT systems. The presence of the magnetic field decreases surface dose due to the elimination of electron contamination [4]. However, electrons affected by the magnetic field can also contribute to out-of-field doses, delivering unwarranted dose to nearby structures [5]. Most notably, traveling electrons, generated within the body by the irradiating beam, are redirected by the Lorentz’s force from a perpendicular magnetic field. In homogeneous tissue, the Lorentz force alters the point spread kernel to be asymmetrical, resulting in an asymmetrical penumbra. At tissue-to-air-interfaces, some electrons are subjected to the electron return effect (ERE), in which the Lorentz force redirects the electrons back upstream to the tissue, resulting in increased tissue dose deposition and potentially treatment hotspots. Therefore, the ERE also creates treatment cold spots at latter air-tissue interfaces due to less tissue dose deposition downstream [4,6,7].

Monte Carlo modeling has been verified, using EBT3 film and thermoluminescent dosimeter measurements in water-based phantoms, to accurately simulate the dosimetric impact of 0.35 T and 1.5 T magnetic fields on air-tissue interfaces and is used clinically in MRgRT treatment planning systems (TPS) for accurate dose calculation [5,8,9,10]. Okamoto et al. and Cusumano et al. showed 0.35 T 60Co Monte Carlo simulations on a water phantom with air gaps was able to achieve a gamma index (dose difference threshold of 3%/ distance to agreement threshold of 3 mm) passing rate greater than 95% [4,6]. Similarly, Shortall et al. showed 1.5 T 7 MV photon Monte Carlo simulation on a polymethyl methacrylate-air phantom was able to achieve gamma index (3%/3 mm) passing rate greater than 95% [11]. In order for Monte Carlo simulation to accurately calculate the dosimetric perturbation caused by the air cavities, accurate electron density needs to be obtained, which can be directly converted from Hounsfield Units (HU) on computed tomography (CT) images. However, most clinical MRgRT treatment planning is based on MR images where electron density is usually obtained by either image registration to CT or bulk density assignment.

A widely adopted MRgRT treatment planning workflow, using the 0.35 T ViewRay MRIdian 6 MV LINAC (ViewRay Inc., Oakwood Village, OH, USA), starts with acquiring both simulation CT and MR images, where the MR image is used as the planning image. Electron density, used in ViewRay TPS Monte Carlo dose calculation, is obtained by deformable registration of the simulation CT to the planning MR. Furthermore, prior to each treatment fraction, a daily MR image is acquired for patient setup and online adaptive therapy can also be considered based on the daily anatomical changes. If online adaptive is deemed necessary, planning contours will be updated and the simulation CT will be re-registered to the daily MRI to provide electron density for adaptive plan recalculation or re-optimization.

In addition to patient daily anatomic changes, air cavities in the gastrointestinal tract can vary greatly between initial simulation and later treatment delivery days. Due to the ERE, daily air cavity variations can potentially lead to deviations of the delivered dose from the planning distribution for abdominal and pelvic treatments [12,13,14]. Uilkema et al. generated 1.5 T 6-MV intensity-modulated radiation therapy (IMRT) plans on 10 rectal cancer patients planning CTs with real air cavities and showed a maximum dosimetric endpoint difference between 1.5 T and no magnetic field of 3% [15]. Scripes et al. generated 1.5 T 7-MV IMRT plans on four rectal cancer patient planning CTs with artificial air cavities and showed planning target volume (PTV) hot spot dose and size increased with increasing air cavity size [16]. PTV hot spot dose increased as much as 10% for 5 cm air cavity relative to no air cavity.

Although MRI-CT deformable registration is fast and generally considered acceptable in obtaining electron densities for soft-tissues and bones, the uncertainty associated with air cavities and their impact on dose calculation is unknown. Furthermore, some adaptive workflows use bulk average electron density assignment on the daily MRI structure from the corresponding simulation CT structure, ignoring structure inhomogeneities and air cavities [17]. As a result, depending on disappearing or appearing daily air cavities, the electron density of the daily structure will be systematically increased or decreased, resulting in inaccurate dose calculations [18,19].

Currently, due to time constraints, daily air cavity contouring and electron density correction is not routinely considered in pelvic MRgART to specifically address its variation and dosimetric uncertainty. Therefore, in this study, the dosimetric impact of air cavities, in a cohort of prostate bed patients, after radical prostatectomy, treated with SBRT on our 0.35 T MRgRT LINAC system, was retrospectively evaluated. Similar to prior air cavity studies, artificial air cavities are generated in a prostate bed patient planning CT [15,16]. However, patient dose will be influenced by a 0.35 T magnetic field rather than a 1.5 T one. Although the magnitude of the ERE is expected to be less with 0.35 T than 1.5 T (weaker Lorentz force), the dosimetric difference has not been quantified. Additionally, dosimetric impact of artificial and real air cavities in prostate bed daily MRI is evaluated, where electron density is obtained from deformable registration with simulation CT. Furthermore, dosimetry between MRgART air cavity electron density corrected and non-corrected (only deformable registration) is compared.

## 2. Materials and Methods

### 2.1. Dosimetric Impact of Artificial Air Cativities in Prostate Bed Patient Planning CT on a 0.35 T MRgRT System

A prostate bed patient (Patient A), with negligible planning CT air cavity, was selected. Artificial tubular air cavities (diameter 0–3 cm, 0.5 cm increments) were generated along the patient’s planning CT rectum in MIM Software (Cleaveland, OH, USA) and bulked assigned air density in ViewRay TPS. Figure 1 shows Patient A’s electron density maps with planning CT contours and different size artificial tubular air cavities.

Three 0.35 T treatment plans were generated for each planning CT air cavity size using ViewRay TPS: (1) single anterior–posterior (AP) beam, (2) opposing AP–PA beams, and (3) clinical IMRT with 15 beams. For the single AP plan, a 5 cm × 5 cm AP beam was used to deliver 5 Gy to the isocenter (center of PTV). Similarly, for the opposing AP–PA plan, two equally weighted 5 cm × 5 cm AP and PA beams were used to deliver 5 Gy to the isocenter (2.5 Gy each). For both plans, dose was calculated on the CT without air cavity (0 cm) using grid resolution of 0.3 cm and magnetic field effect on. Afterwards, both plans were forward calculated on the CTs with various air cavity sizes (0.5–3 cm) using dose uncertainty of 1%, respectively.

For the clinical IMRT plan, 15 equally arced beams were placed around the isocenter (PTV) and 32 Gy was prescribed to 95% of the PTV (5 fraction (Fx); 6.4 Gy/Fx). Dose was optimized on the CT without air cavity using clinical constraints (Table 1), grid resolution of 0.2 cm, and magnetic field effect on. Similarly, the optimized plan was forward calculated on the CTs with various air cavity sizes (0.5–3 cm) using dose uncertainty of 1%. 

### 2.2. Dosimetric Impact of Artificial Cavities Air Cavities in Prostate Bed Patient Daily MRI, with Electron Density Obtained from Deformable Registraion to Simulation CT, on a 0.35 T MRgRT System

A different prostate bed patient (Patient B), with negligible simulation CT and daily MRI air cavity, was selected. Artificial tubular air cavities (diameter 0–3 cm, 0.5 cm increments) were generated along the patient’s daily MRI rectum and masked with a signal intensity of 0 in MIM Software, simulating daily appearing air cavities that are not present during the initial CT simulation. In ViewRay TPS, the daily MRIs with various air cavity sizes were each deformably registered to the simulation CT for electron density mapping and treatment planning. Figure 2 shows Patient B’s daily MRI with different size artificial tubular air cavities and corresponding electron density map from deformable registration with simulation CT. Notably, deformed electron density maps were unable to recreate daily MRI air cavities, resulting in inaccurate daily MRI electron density maps for dose calculation.

Two 0.35 T MRgART treatment plans were generated for each daily MRI air cavity size using ViewRay TPS: Plan 1—‘Deformed’ and Plan 2—‘Override’. In the ‘Deformed’ plan, a clinical IMRT plan with 15 equally arced beams around the daily PTV was generated on the daily MRI, with electron density map obtained from the registered simulation CT, to deliver 32 Gy to 95% of the PTV. Dose was optimized using the same constraints as in Table 1, grid resolution of 0.3 cm, and magnetic field effect on. In the ‘Override’ plan, daily MRI air cavities were corrected and bulk assigned air density on the registered simulation CT’s electron density map. Afterwards, the ‘Deformed’ plan was forward calculated on the daily MRI with a dose uncertainty of 1%.

### 2.3. Dosimetric Impact of Real Air Cavities in Prostate Bed Patient Daily MRI, with Electron Density Obtained from Deformable Registraion to Simulation CT, on a 0.35 T MRgRT System

Five prostate bed patients (Patient 1–Patient 5) with non-negligible daily MRI air cavities were selected. Prior to simulation and each daily MRI fraction, patients were instructed to follow institutional bladder and rectum filling protocol. After simulation, an initial clinical 0.35 T MRgRT IMRT plan was generated on the planning MRI with electron density obtained from deformable registration to simulation CT. Initial clinical IMRT plans for the five prostate bed patients used 13–17 beams arced around the PTV to deliver 32 Gy to 95% of the PTV. Dose was optimized using the same constraints as in Table 1, grid resolution of 0.3 cm, and magnetic field effect on.

For each patient, two fractions (Fraction A and Fraction B), with the largest daily MRI air cavities, were selected to simulate the adaptive planning process. Planning MRI contours were updated to the daily MRI anatomy by a resident radiation oncologist and reviewed by the attending physician in MIM Software. Afterwards, in ViewRay TPS, each daily fraction MRI was deformably registered to its simulation CT.

Two 0.35 T MRgART treatment plans were predicted for each fraction in ViewRay TPS: Plan 1—‘Deformed’ and Plan 2—‘Override’. In the ‘Deformed’ plan, the initial clinical plan was forward calculated on the daily MRI with electron density map obtained from the registered simulation CT. In the ‘Override’ plan, simulation CT and daily MRI air cavities were manually corrected and bulk assigned water and air density on the registered simulation CT’s electron density map, respectively. Afterwards, the initial clinical plan was forward calculated. For ‘Deformed’ and ‘Override’ plans, the initial clinical plan was forward calculated with dose uncertainty of 1%. Figure 3 shows Patient 3 Fraction A’s daily MRI, corresponding deformed simulation CT, deformed electron density map (‘Deformed’), and corrected daily air cavity electron density map (‘Override’). Daily MRI and deformed simulation CT display significantly different air cavities. White arrows indicate workflow for obtaining daily air cavity corrected electron density map. After daily air cavity correction, the ‘Override’ electron density map more accurately represents the daily MRI’s electron density.

### 2.4. Dosimetric Evaluation

For the artificial planning CT air cavities (Section 2.1), prostate bed (Patient A) AP, AP–PA and clinical IMRT plan dose difference maps, for each air cavity size (0.5–3 cm), relative to no air cavity (0 cm), were calculated and normalized to each respective plan’s prescribed PTV dose. Planning CT target/organ mean and max dose difference were evaluated. Additionally, AP dose-depth curves along the isocenter were plotted for all air cavity sizes and plans.

For the artificial daily MRI air cavities (Section 2.2), prostate bed (Patient B) dose difference maps between ‘Deformed’ and ‘Override’ plans for each air cavity size (0.5–3 cm), were calculated. Similarly, daily MRI target/organ mean and max dose difference were evaluated.

For the real daily MRI air cavities (Section 2.3), prostate bed (Patients 1–5) air cavity volume changes, between simulation CT and daily MRIs (Fractions A and B), were calculated. Furthermore, clinical significance of MRgART air cavity electron density correction was evaluated based on the clinical constraint changes. MRgART air cavity electron density correction was considered significant if the ‘Deformed’ plan passed the clinical constraint, but the ‘Override’ plan did not, or vice-versa.

## 3. Results

### 3.1. Dosimetric Impact of Artificial Air Cativities in Prostate Bed Patient Planning CT on a 0.35 T MRgRT System

Figure 4 shows prostate bed Patient A’s AP plan dose and dose difference maps for 0, 1, 2, and 3 cm artificial planning CT air cavities. Similarly, Figure 5 and Figure 6 shows the AP–PA and clinical IMRT plan dose and dose difference maps. Figure 7 shows the AP dose-depth curve, along the isocenter, for varying air cavity sizes with AP, AP–PA, and clinical IMRT plans. For the AP and AP–PA plan, hot and cold spots, within the rectum, significantly increased in magnitude and size, relative to no air cavity, with increasing air cavity size due to the ERE, where the 3 cm air cavity resulted in a 20%/17% and 13%/13% increase to the rectum/rectal wall max dose, respectively. ERE, for the AP–PA and clinical plan, were dominated by the posterior beams due to less attenuation on the posterior end of the target. As a result, the hot and cold spots appear rotated relative to the AP plan. Overall, the magnitude and size of the hot spot decreased when multiple beams were used as the dose was more uniformly spread across the body and ERE was averaged-out by opposing beams. Clinical IMRT dose, with multiple optimized equally arced beams, was robust to air cavity size and deviations remained within dose calculation uncertainty (1%). The complete target/organ max and mean dosimetric change for each air cavity size and plan can be found in Appendix A (Table A1).

### 3.2. Dosimetric Impact of Artificial Cavities Air Cavities in Prostate Bed Patient Daily MRI, with Electron Density Obtained from Deformable Registraion to Simulation CT, on a 0.35 T MRgRT System

As shown in Figure 2, daily MRI deformable registration with simulation CT is unable to create daily MRI appearing air cavities on the resulting electron density map. Figure 8 shows prostate bed Patient B’s ‘Deformed’ and ‘Override’ plans’ dose and dose difference maps for 1, 2, and 3 cm artificial daily MRI air cavities. Target/organ mean and max dose difference was relatively small (<4%) even though deformable registration was not able to attain electron density for the air cavities. Furthermore, no noticeable dose trend as a function of air cavity size was observed.

### 3.3. Dosimetric Impact of Real Air Cavities in Prostate Bed Patient daily MRI, with Electron Density Obtained from Deformable Registraion to Simulation CT, on a 0.35 T MRgRT System

For prostate bed Patients 1–5 Fraction A and B, the mean (range) air cavity size change between daily MRI and simulation CT was 5.69 ± 10.07 cc (−7.8 to + 22.9 cc), where positive and negative air cavity change represent increasing and decreasing air volume.

Table 2 shows the mean constraint change between ‘Deformed’ and ‘Override’ plans for all patient fractions based on respective daily MRI with real air cavities. The complete daily constraint and change for each plan, patient and fraction can be found in Appendix A (Table A2). The dosimetric difference between ‘Deformed’ and ‘Override’ plans did not lead to change in clinical adaptive decision except for two fractions (Patient 3 Fraction A and Patient 4 Fraction B). In these fractions, the ‘Override’ plan, with air cavity electron density correction, indicated that the bladder D_max_ dose and rectum V35.7 exceeded the constraint, while the ‘Deformed’ plan, without correction, showed acceptable dose, although the absolute difference was only 0.3 Gy and 0.03 cc, respectively. Overall, clinical constraint changes between ‘Deformed’ and ‘Override’ plans were minimal.

## 4. Discussion

This study evaluated the dosimetric impact of air cavities in prostate bed patients’ rectums on a 0.35 T MRgRT system. Artificial tubular air cavities of various sizes (0–3 cm) were generated on a prostate bed patient’s planning CT and three different plans (AP, AP–PA, clinical IMRT), were retrospectively evaluated. In the AP and AP–PA plans, the rectum and rectal wall hotspot magnitude and size increased with increasing air cavity size. The AP–PA plan hot spot dose was lower than the AP plan, and was further reduced using clinical IMRT. The use of opposing beams counter-balance and average-out ERE from each respective beam, reducing the magnitude and size of hot spots due to air cavities [12,14]. Comparing our clinical IMRT results with a previous artificial air cavity clinical IMRT study [16], using a 3 cm air cavity, the maximum rectum hot spot change (relative to 0 cm) was expectedly lower for 0.35 T (1%) than 1.5 T (6%).

In our second experiment, artificial tubular air cavities were generated on a prostate bed patient’s daily MRI, and electron density information was obtained using deformable registration with simulation CT. Deformed electron density maps were unable to reproduce daily MRI air cavities; however, the dose differences between air cavity electron density corrected (‘Override’) and uncorrected (‘Deformed’) clinical IMRT plans were minimal (< 4%) due to beam averaging and low magnetic field strength. Therefore, prostate bed dose calculation, based on electron density directly derived from MR-CT deformable registration, is acceptable for various rectal air cavity sizes under a low strength magnetic field.

Lastly, MRgART air cavity electron density correction was evaluated on prostate bed patients’ daily MRI with real air cavities. Daily clinical constraint changes between ‘Deformed’ and ‘Override’ plan were minimal, although large variations of air cavity size were observed. MRgART air cavity electron density correction was considered clinically significant if: (1) ‘Deformed’ plan failed clinical constraint while ‘Override’ plan passed; or (2) ‘Deformed’ plan passed clinical constraint while ‘Override’ plan failed. In Scenario 1, ‘Deformed’ plan constraints failing would suggest the need for further plan adaptation; however, if the correct daily air electron density was used, plan adjustment would not be necessary, reducing treatment time. In Scenario 2, ‘Deformed’ plan constraints passing would suggest plan adjustment is not needed based on deformably registered CT only; however, further plan adaptation is actually necessary when the correct air cavity electron density is used. Of the 10 real patients’ air cavity fractions, two fractions followed Scenario 2, indicating air cavity electron density correction is necessary to improve prostate bed IMRT constraints. Despite this, the absolute difference between ‘Deformed’ and ‘Override’ plans were small in these cases. Furthermore, air cavity electron density correction is a time-intensive procedure that requires daily air cavity segmentation and plan re-calculation or re-optimization. As a result, MRgART, based on daily anatomical change, is sufficient [20], and the need for additional air cavity electron density correction to detect small dosimetric change is not warranted at the cost of significantly longer treatment times.

This study has several limitations. First, our real air cavity patient cohort size was small (5 patients, 2 fractions each). A larger patient cohort can include patients with different size and shape air cavities. However, even with the current patient cohort, with a large range of air cavity sizes, the dosimetric change from air cavity electron density correction is negligible. Second, this study used ViewRay, vendor-specific, deformable registration to obtain electron density maps for daily MRIs. Deformable registration algorithms, using machine learning, can be used between daily MRI and simulation CT for potentially more accurate electron density mapping [21]. Despite potential improvements in electron density accuracy, manual air cavity electron density correction used in this study resulted small dosimetric differences with little to no clinical significance. Lastly, this study used Viewray, vendor-specific, Monte Carlo dose calculation, which may use underlying model assumptions to accelerate dose calculation and thus is not generalizable to other systems. However, Khan et al. developed a general vendor-independent Monte Carlo 0.35 T/6 MV MR-LINAC model using GEANT4 code and was able to show good agreement with ViewRay TPS results [22]. Therefore, our results, using ViewRay TPS Monte Carlo dose calculation, is a reasonable estimation of the dosimetric impact of air cavities in prostate bed patients on a 0.35 T MRgRT system.

## 5. Conclusions

Clinical 0.35 T IMRT prostate bed plans are dosimetrically robust to rectum air cavity size as multiple beam angles are used and optimized, resulting in dosimetric deviations, stemming from the ERE, to be averaged out. Moreover, 0.35 T ERE dosimetric deviations are smaller than 1.5 T. Despite strict bladder and rectum filling protocol, air cavity size change between simulation and daily treatment are still present. Daily MRI electron density map, obtained from deformable registration with simulation CT, is unable to produce accurate daily air cavity electron density. However, 0.35 T MRgART plan recalculation or re-optimization, with air cavity electron density correction, shows small and clinically insignificant dosimetric change relative to uncorrected deformed electron density plans. Therefore, MRgART air cavity electron density correction is not warranted for prostate bed patients treated with clinical IMRT on low-field MRgRT systems.

## Figures and Tables

**Figure 1 jcm-11-00364-f001:**
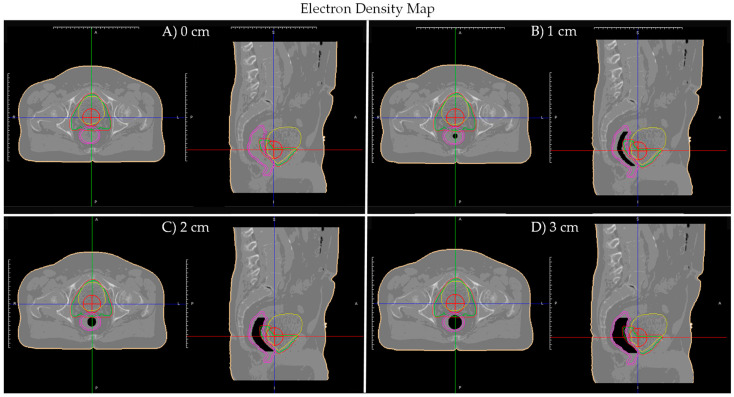
Patient A’s planning CT electron density maps with artificial tubular air cavities of diameter: (**A**) 0 cm; (**B**) 1 cm; (**C**) 2 cm; and (**D**) 3 cm. Planning CT contours displayed: Red—planning target volume (PTV); Green—clinical target volume (CTV); Yellow—bladder; Pink—rectum/rectal wall; Red cross—isocenter.

**Figure 2 jcm-11-00364-f002:**
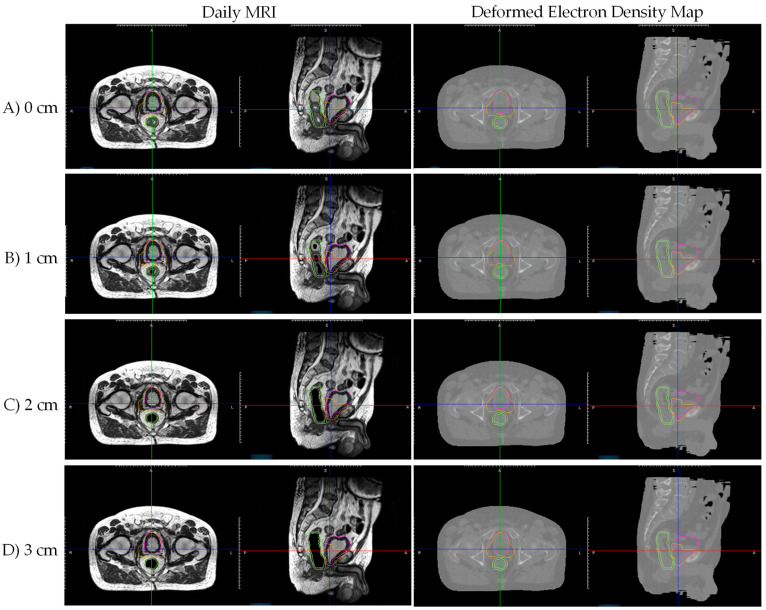
Patient B’s daily MRI with artificial tubular air cavities of diameter (**A**) 0 cm, (**B**) 1 cm, (**C**) 2 cm, and (**D**) 3 cm, and corresponding electron density map from deformable registration with simulation CT. Notably, deformed electron density maps were unable to recreate daily MRI air cavities, resulting in inaccurate daily MRI electron density maps. Daily MRI contours displayed: Pink—bladder; Orange—PTV; Green—rectum/rectal wall.

**Figure 3 jcm-11-00364-f003:**
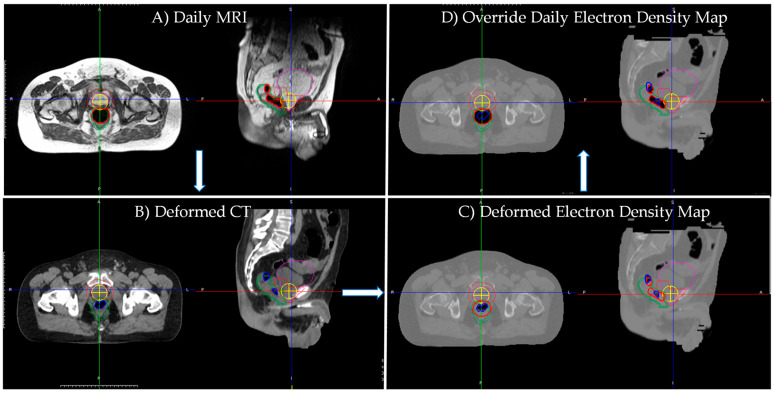
Patient 3 Fraction A’s (**A**) daily MRI, (**B**) corresponding deformed simulation CT, (**C**) deformed planning electron density map (‘Deformed’) and (**D**) corrected daily air cavity electron density map (‘Override’). Daily MRI and deformed simulation CT display significantly different air cavities. White arrows indicate workflow for obtaining daily air cavity corrected electron density map. After daily air cavity correction, the ‘Override’ electron density map more accurately represents the daily MRI’s electron density. Daily MRI contours displayed: Pink—bladder; Orange—PTV; Green—rectum/rectal Wall; Red—daily MRI air cavity; Yellow cross—isocenter. Simulation CT contour displayed: Blue—simulation CT air cavity.

**Figure 4 jcm-11-00364-f004:**
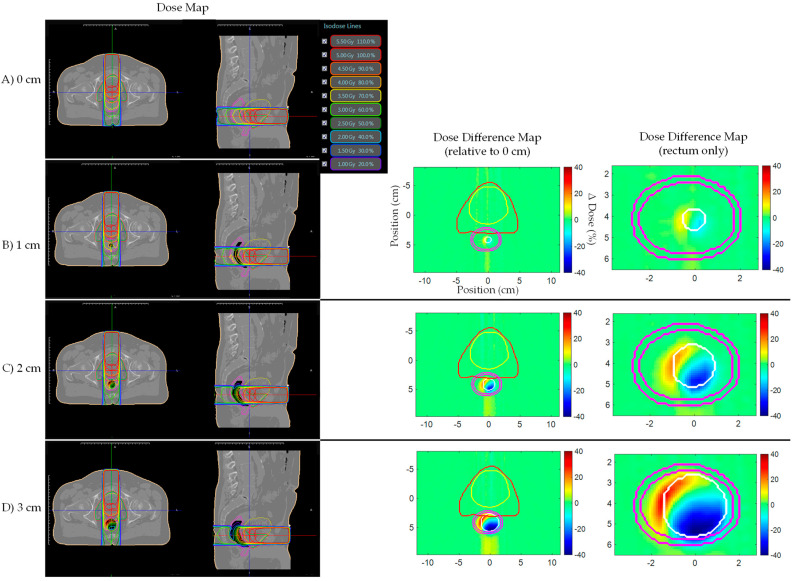
Prostate bed Patient A’s AP plan dose maps (overlaid on planning CT electron density map) and dose difference (ΔDose (%)) maps for (**A**) 0, (**B**) 1, (**C**) 2, and (**D**) 3 cm artificial planning CT air cavity Rectum dose difference maps show hot and cold spots within the rectum increased in magnitude and size with increasing air cavity size due to the electron return effect (ERE). Dose difference map planning CT contours displayed: Red—PTV; Yellow—bladder; Pink—rectum; White—artificial air cavity.

**Figure 5 jcm-11-00364-f005:**
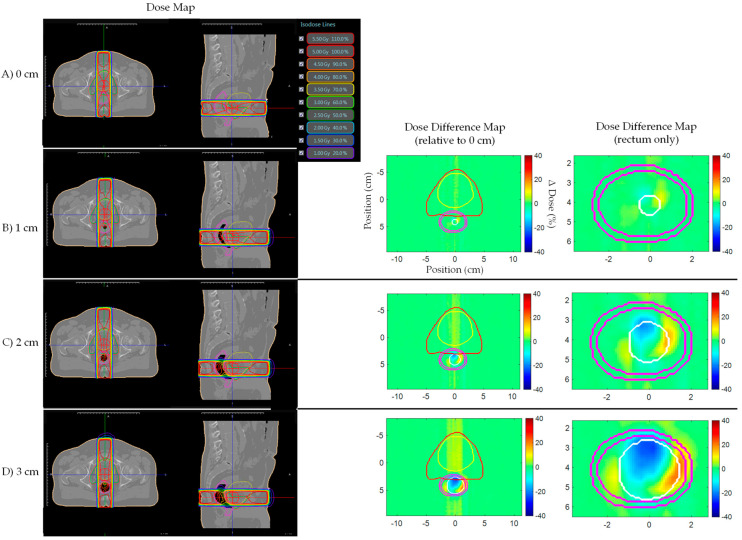
Prostate bed Patient A’s AP–PA plan dose maps (overlaid on planning CT electron density map) and dose difference (ΔDose (%)) maps for (**A**) 0, (**B**) 1, (**C**) 2, and (**D**) 3 cm artificial planning CT air cavity. Rectum dose difference maps show hot and cold spots within the rectum increased in magnitude and size with increasing air cavity size due to the electron return effect (ERE). Dose difference map planning CT contours displayed: Red—PTV; Yellow—bladder; Pink—rectum; White—artificial air cavity.

**Figure 6 jcm-11-00364-f006:**
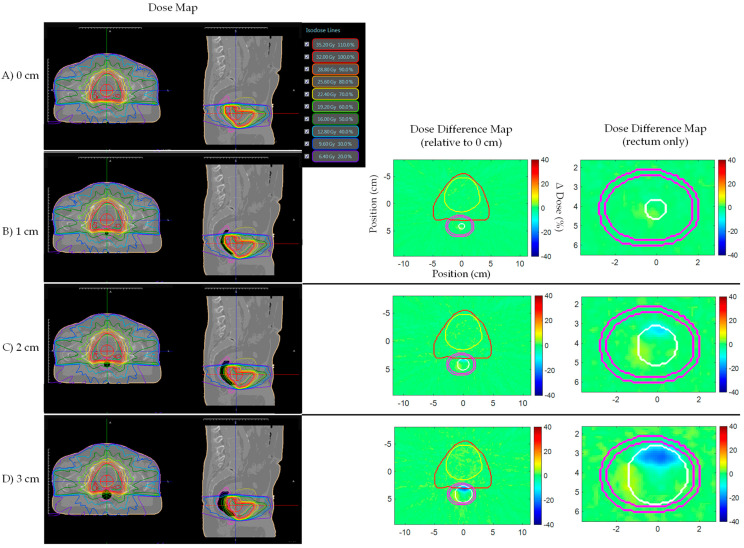
Prostate bed Patient A’s clinical IMRT plan dose maps (overlaid on planning CT electron density map) and dose difference (ΔDose(%)) maps for (**A**) 0, (**B**) 1, (**C**) 2, and (**D**) 3 cm artificial planning CT air cavity. Rectum dose difference maps show little to no dose change within the rectum for different air cavity sizes due to dose being uniformly spread and the electron return effect (ERE) being averaged-out by multiple opposing beams. Dose difference map planning CT contours displayed: Red—PTV; Yellow—bladder; Pink—rectum; White—artificial air cavity.

**Figure 7 jcm-11-00364-f007:**
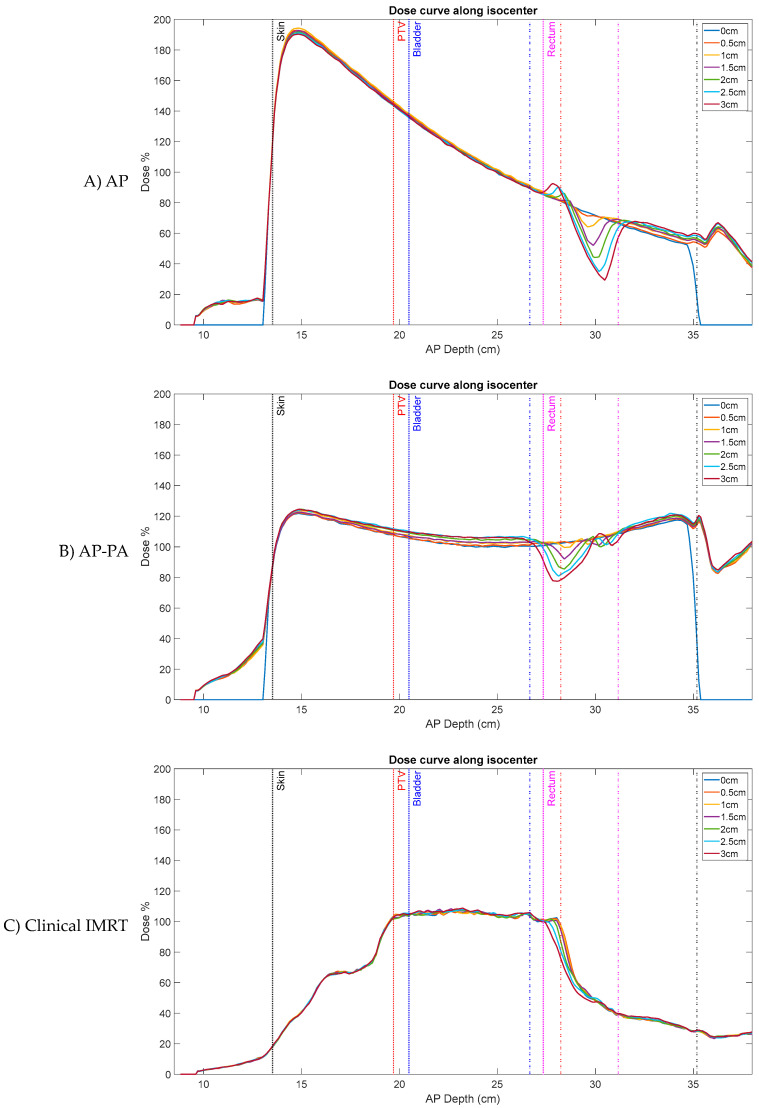
Prostate bed Patient A’s AP dose-depth curve, along the isocenter, for varying air cavity sizes with (**A**) AP, (**B**) AP–PA, and (**C**) clinical IMRT plans (normalized by respective plan’s prescribed PTV dose). Clinical IMRT dose is more robust to air cavity sizes than AP and AP–PA plans and shows little deviation. Solid lines—planning CT target/organ start, dashed lines—planning CT target/organ end.

**Figure 8 jcm-11-00364-f008:**
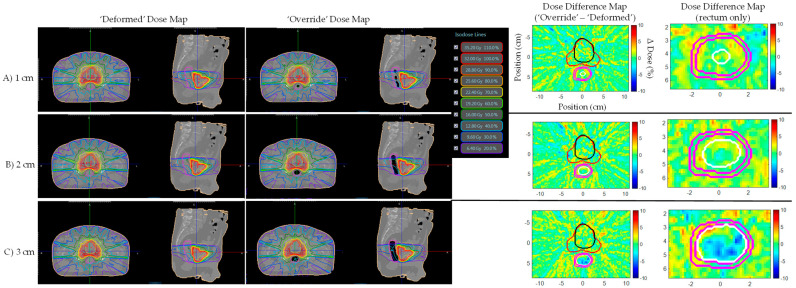
Prostate bed Patient B’s ‘Deformed’ and ‘Override’ plan dose and dose difference maps for (**A**) 1, (**B**) 2, and (**C**) 3 cm artificial daily MRI air cavities. Target/organ mean and max dose difference was relatively small (<4%) despite air cavity correction in the ‘Override’ plan. Dose difference map daily MRI contours displayed: Red—PTV, Black—bladder, Pink—rectum, White—artificial air cavity.

**Table 1 jcm-11-00364-t001:** Clinical IMRT constraints for prostate bed patients.

Constraint	
PTV V32	≥95%
Bladder Max	≤35.7 Gy
Bladder V35.7	≤0.03 cc
Bladder V32.5	≤35%
Rectum Wall V24	≤50%
Rectum Max	≤35.7 Gy
Rectum V35.7	≤0.03 cc
Rectum V33.75	≤25%
Rectum V32.5	≤30%
Rectum V27.5	≤45%

**Table 2 jcm-11-00364-t002:** Mean constraint change between ‘Deformed’ and ‘Override’ plans for all patient (Patient 1–5) fractions (Fraction A and B) based on respective daily MRI with real air cavities.

Constraint	Change
PTV V32	−0.27 ± 0.27%
Bladder Max	−0.01 ± 0.24 Gy
Bladder V35.7	0.04 ± 0.12 cc
Bladder V32.5	0.09 ± 0.87%
Rectum Wall V24	−0.25 ± 0.37%
Rectum Max	−0.07 ± 0.50 Gy
Rectum V35.7	0.00 ± 0.01 cc
Rectum V33.75	−0.17 ± 0.37%
Rectum V32.5	−0.46 ± 0.66%
Rectum V27.5	−0.72 ± 1.01%

## Data Availability

The data in this study are not publicly available.

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
