# Peer review of "Dosimetric Effects of Air Cavities for MRI-Guided Online Adaptive Radiation Therapy (MRgART) of Prostate Bed after Radical Prostatectomy"

_jcm, 2022, doi:10.3390/jcm11020364_

Round 1

Reviewer 1 Report

General comments: 

The paper is very interesting, but it is so hard to read, that the reader who is not fully into the topic is unable to get through the study. It should be more reader-friendly written and more explanation should be included. For instance, the aim of the study is not obvious and the problem that the authors want to study is nowhere discussed. The authors should extend the introduction and discuss the study more widely. Also, the text is very poorly formatted. Otherwise, a nice paper.

Major comments:

At first - the abstract is too long for the standards of MDPI - it has to be shorter and focus on bullet points of the study.

Second, the introduction lacks sufficient background for researchers not familiar with the topic of the study. JCM is also a base of knowledge for clinicians that are not as fluent in physical bases as the authors of the study. Thus I recommend extending the section and including short reasoning why the authors chose such parameters for the study. 

As for the Materials and Methods section, the MRI scans should be of better quality - from these photographs, it is hard to follow the text. Moreover, some abbreviations, like CT are not explained in the text. Most are self-explanatory, but some remain a challenge for the reader. Also, the authors have a very confusing numeration of figures in the text - there are two "figure 2"s and I don't feel the references to them in the text are adequate... should be corrected. Also, layout figures should be divided into sections (A, B, C...) and each of them should be fully explained.

References section should be prepared and formated with some professional software according to formating standards of MDPI.

The study is interesting, but the presentation is very poor... 

Reviewer 2 Report

The manuscript with the title “Dosimetric Effects of Air Cavities for MRI-guided Online Adaptive Radiation Therapy (MRgART) of Prostate Bed after Radical Prostatectomy” is a well-structured and informative article. The authors evaluate the dosimetric impact of air cavity electron density correction for 0.35T MR-guided Online Adaptive Radiation Therapy (MRgART) of prostate bed patients after radical prostatectomy. The results of the study are significant for the relative field and numerous comparisons are made. I have  two minor comments, as  explained below:

  • The Abstract of the article is extensive and may confuse the reader. Moreover, it does not attract the attention of the reader. Therefore, it is highly recommended to present a more concise Abstract.
  • The quality of the figures could be significantly improved, as in their current form they are not clear.

Reviewer 3 Report

I have read with great interest this manuscript concerning the impact of air cavities for MR-guided adaptive post-operative RT for prostate cancer. The data are properly collected and presented. The manuscript is well written and no major concerns are raised. I recommend it for publication, but I only suggest to include among the references the following article and to briefly mention it in the Discussion: 

- Daily dosimetric variation between image-guided volumetric modulated arc radiotherapy and MR-guided daily adaptive radiotherapy for prostate cancer stereotactic body radiotherapy. Acta Oncol. 2021 Feb;60(2):215-221. doi: 10.1080/0284186X.2020.1821090. 

Reviewer 4 Report

The authors evaluated the dosimetric impact of rectal air cavity electron density correction for 0.35T MR-guided Online Adaptive Radiation Therapy (MRgART) of prostate bed patients after radical prostatectomy. The limitations of the study were reported. The authors conclude that MRgART rectal air cavity electron density correction is not warranted for prostate bed patients treated with clinical IMRT on low-field MRgRT systems. The manuscript is well conducted and I have no other changes to add.

Round 2

Reviewer 1 Report

The paper was heavily revised by the Authors and in my opinion.

Author Response

Thank you very much for your review.